# Strategies for improved endothelial cell adhesion in microphysiological vascular model systems

**Jingyi Zhu**[1], **Halie L. Hotchkiss**[2], **Kevin L. Shores**[1], **George A. Truskey**[1*],
**Stacey A. Maskarinec**[2]

**1** Department of Biomedical Engineering, Duke University, Durham, North Carolina, United States of America, **2** Division of Infectious Diseases, Duke University Health System, Durham, North Carolina, United States of America

* george.truskey@duke.edu

## Abstract

Human tissue-engineered blood vessels (TEBVs) have been applied as model systems to study a wide range of vascular diseases including Hutchinson-Gilford Progeria Syndrome and early atherosclerosis. Central to the utility of TEBVs as an *in vitro* blood vessel model is the maintenance of a functional endothelium under physiologically relevant shear stresses. Establishing and maintaining a confluent endothelial monolayer is challenging. In this protocol, we outline an optimized procedure for the endothelialization of TEBVs. We optimized the following key conditions affecting endothelial cell (EC) adherence in the vessel: EC seeding density, rotation time, and the application of perfusion. This protocol results in TEBVs with sustained EC luminal coverage that demonstrate alignment in the direction of applied flow and responsiveness to inflammatory stimuli. To facilitate rapid screening of EC coverage during the fabrication and perfusion steps, we re-designed TEBV chambers to include a viewing window that allows for efficient monitoring and assessment of the endothelialization process using fluorescence microscopy. By identifying key factors that affect EC attachment in TEBVs, this protocol may serve as a valuable resource for researchers seeking to achieve successful endothelialization of engineered blood vessel constructs.

## Introduction

Vascular diseases, which encompass a wide range of conditions from atherosclerosis to inflammatory vasculitis, continue to have a tremendous impact on global health [1–3]. To better understand disease progression and identify novel therapeutic targets, reliable and accessible model systems that recapitulate the spatial distribution of cells within blood vessels are needed. Tissue-engineered blood vessels (TEBVs) using human cells can replicate both the structural and functional properties of native blood vessels and represent a useful model platform for this purpose [4,5]. Unlike 2D

**Data availability statement:** All raw data required to replicate the results of the study are included. The supporting the figures is supplied as a supplementary file entitled "Raw Data".

**Funding:** This research was funded by grants from the National Institutes of Health: K23 HL159275 to S.M. and R01 HL138252 to G.T. (https://www.nih.gov/) an American Heart Association Collaborative Sciences Award 957133 to S.M. and G.T. (https://www.heart.org/), and an American Heart Association predoctoral fellowship (388000147) to J.Z.(https://www.heart.org/). The funders did not have a role in study design, data collection and analysis, decision to publish, or preparation of the manuscript.

**Competing interests:** The authors have declared that no competing interests exist.

systems, TEBVs offer the unique advantage of systematically characterizing endothelial cell (EC) responses to various pathological stimuli while reproducing spatial complexity, maintaining flow conditions, modeling leukocyte adhesion and infiltration, and facilitating the study of cell-cell interactions. Recent studies have demonstrated new insights into EC development, plasticity, and heterogeneity, and can be further explored in representative model systems to elucidate factors involved in disease pathogenesis, ultimately contributing to improved clinical outcomes [6–8].

ECs form the innermost layer of all blood vessels, and their function is to ensure patency, inhibit thrombosis, and mediate processes including solute permeability, vasomotor tone, and immune cell recruitment or inhibition [9–11]. Since alterations in EC response contribute to disease pathogenesis, the establishment of a robust EC monolayer is crucial for TEBV function. Despite their usefulness in studying vascular diseases, a major challenge in TEBV generation is forming and maintaining an intact monolayer of ECs. Seeding conditions must enable uniform attachment and spreading of ECs. When exposed to shear stress from flowing media, cells that have not adhered to the substrate are prone to detachment. Additional downstream processes and bubbles in the perfusion system can dislodge these cells and may further exacerbate EC loss.

To address these challenges, multiple strategies have been utilized to enhance EC attachment in tissue-engineered vessel constructs, with particular emphasis on endothelialization methods and the application of shear stress. For instance, decellularized polyglycolic acid (PGA)-based mesh scaffolds seeded with either porcine peripheral blood endothelial progenitor cells (EPCs) or ECs showed improved attachment when the vessel was rotated 90° every 30 minutes for 2 hours, followed by a 2-hour static incubation, then a gradual application of shear stress from 1 to 15 dyne/cm² over 36 hours [12]. Similarly, human saphenous vein endothelial cells achieved more attachment to collagen-based scaffolds after being exposed to a rotation protocol involving 20 seconds of clockwise rotation followed by 30 seconds of counterclockwise rotation, with a 20-minute holding phase between each cycle, over a 7-hour period. The application of shear stress, starting 4 hours after seeding and gradually increasing from 10 to 20 dyne/cm² over 18 hours, further enhanced endothelialization [13]. While these approaches result in improved endothelialization of tissue engineered constructs, they often require access to specialized bioreactor equipment and generate a limited number of vessel replicates.

In our previously published work [14–16], we implemented protocols for endothelializing TEBVs using a $1 \times 10^6$/mL EC suspension with rotation at 10 revolutions per hour for 30 minutes. Additionally, in our protocol paper [4], we modified this approach by seeding 4 TEBVs with a $4–6 \times 10^6$/mL EC suspension. ECs were allowed to adhere statically for at least 30 minutes, followed by rotating the chamber every 15 minutes. However, a comprehensive investigation of EC seeding density, rotation speed, and duration has not yet been conducted. In this study, we addressed this gap by modifying the design of the perfusion chamber and established improved EC retention within a collagen-based scaffold TEBV chamber system (4 vessels per chamber) under physiologically relevant flow conditions. Specifically, we incorporated

a viewing window into our established perfusion chamber [4] and improved the procedural workflow during the initial fabrication and perfusion steps resulting in increased EC attachment.

Establishing a viewing window in the perfusion chamber for real-time microscopic visualization of the experimental process allows investigators to assess EC coverage within TEBVs *in situ* using conventional phase contrast and/or fluorescence microscopy. Overcoming the limitations of the previous chamber, which was restricted to stereoscopic imaging due to the thickness and reduced optical clarity of the polycarbonate chip, this new viewing chamber minimizes the working distance between the microscope objective and the TEBVs by incorporating two recessed areas on the top surface of the perfusion chamber. The innermost recessed area permits the attachment of a standard glass coverslip to enhance image resolution, while the outer recessed area further reduces the working distance and is optimized to enable visibility of all four TEBV lumens during horizontal scanning. In addition, we designed removable polydimethylsiloxane (PDMS)-based clamps to secure TEBVs within the chamber during perfusion, eliminating the need to suture the ends of the TEBVs over stainless steel tubing connected to the flow loop and shortening the time to prepare the TEBVs. Together, these features enable the rapid identification and removal of TEBVs with damaged lumens or compromised EC coverage, minimize downstream complications, and improve reproducibility. We provide detailed schematics for both the chamber and clamp designs (Supporting Information S1 File), all of which were generated using free/open-source software, to facilitate replication and customization by other researchers.

The second aspect of our approach includes a comprehensive workflow to improve EC adherence within the TEBV lumens (Fig 1A). EC retention is related to the conditions during dynamic rotational seeding and the application of shear stress. Optimization of dynamic rotational seeding was determined by examining the impact of varying EC seeding density and the duration of rotation. Perfusion optimization was determined by serially monitoring EC coverage for 1 week following a rapid or gradual onset of shear stress (Fig 1B). The rotational speed used in the experiments was determined by the rotator system. To reduce the speed and improve endothelial cell seeding, we modified an existing rotator by incorporating a commercially available 20:1 worm gear. Although our target range of 18–22 revolutions per hour (rph) was informed by prior work [16–19], limitations of the available rotator prevented further adjustment. Empirical measurements with the modified system showed that one full rotation occurred approximately every 2.5 minutes, corresponding to a speed of 24 rph. We also provide simulations that justify a specific range of rotation speeds. These parameters are critical in establishing TEBVs with robust EC coverage and are applicable to other laboratories engineering similar vascular conduits.

This step-by-step protocol provides detailed instructions for optimized EC seeding densities, rotation duration, and perfusion application. TEBVs generated using this protocol were shown to support a functional endothelium, as evidenced by the formation of cell-cell adhesion junctions, alignment in the direction of flow, and the upregulation of immune cell adhesion molecules following treatment with inflammatory cytokines.

## Methods

As illustrated in Fig 1A, the protocol begins with the preparation of necessary components for TEBV fabrication and perfusion. TEBVs are fabricated by injecting the collagen mixture with human neonatal dermal fibroblasts (hNDFs) into the mold chamber, forming the outer layer of the construct. Following gelation and dehydration, this hNDF layer creates a robust scaffold for endothelialization. HUVECs are then seeded onto the luminal surface of the TEBVs and are subjected to controlled rotation to achieve uniform EC coverage. Once seeded, the TEBVs are integrated into a perfusion system with two interconnected loops. To promote TEBV maturation, the perfusion flow rates are gradually increased to physiological levels. After a 7-day maturation period, the TEBVs can be used for various downstream experiments to validate the functionality and utility of the system. CAD files for the TEBV chambers, 3D printed chambers holder and PDMS clamp mold are provided in Supporting Information S1 File. The detailed steps for material preparation, TEBV fabrication, downstream experiments, and troubleshooting tips described in this study are published on the protocols.io

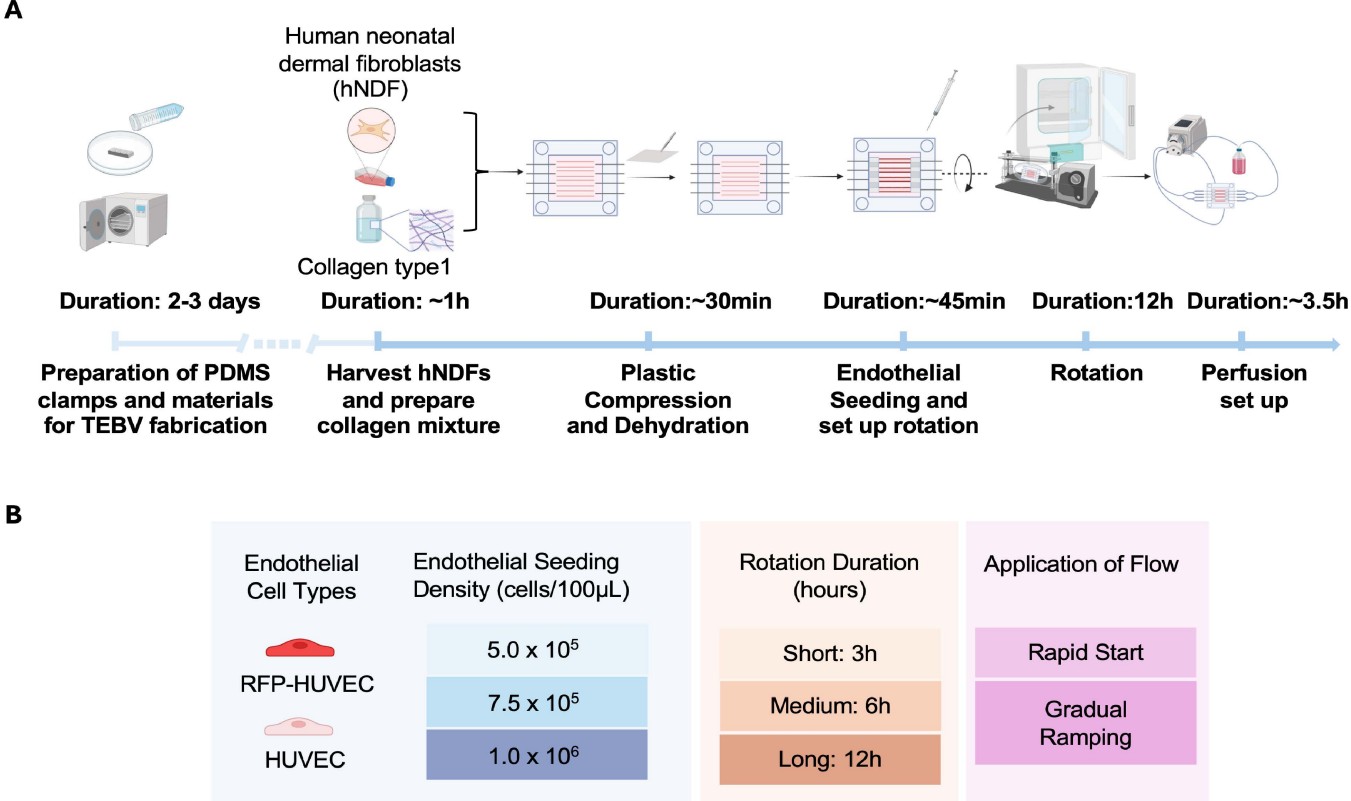

**Fig 1. Overview of tissue-engineered blood vessel (TEBV) fabrication workflow and experimental design. (A)** Fabrication and sterilization of clamps and other necessary materials for TEBV fabrication should be completed prior to the start of the experiment. TEBV fabrication workflow begins with preparation of collagen type I mixture with human neonatal dermal fibroblasts (hNDFs). *In situ* plastic compression and dehydration is performed to remove excessive water content from the TEBVs. Subsequently, endothelial cells (ECs) are injected into the vessel lumen. To facilitate uniform cell adhesion, the chamber undergoes a 12-hour rotation in the incubator. Gradual ramping of perfusion starts the following day. **(B)** Both human umbilical endothelial cells (HUVEC) and red fluorescent protein (RFP)-tagged HUVEC are used in this study. Optimization parameters including seeding density, rotation duration and flow application were explored. Figure created in Biorender (https://BioRender.com).

(dx.doi.org/10.17504/protocols.io.3byl4wpwzvo5/v1) and are provided as a supplementary file with this article (Supporting Information S2 File Protocol). A step-by-step video of the protocol is also provided as Supporting Information (S3 Video).

## Expected results

### A. Viewing chamber enables *in situ* visualization of endothelialization process

Expanding on the design described by Zhang *et al.*[4], the current chamber was re-designed to include a viewing window to support advanced imaging techniques. The viewing window consists of two rectangular, recessed areas: an inner area (17.5 mm × 19 mm) and an outer area (25.5 mm × 35.5 mm) (Fig 2A). The inner recessed area is 0.2 mm deep in the center of the chamber panel. The enlarged area of the inner recess allowed direct visualization of all 4 TEBVs *in situ*, and the depth of the inner recess was maximized in order to decrease the distance between the TEBV lumen and microscope objective. The outer recessed area is 3.5 mm deep to the center of the chamber panel which is sufficient to fit the front lens assembly housing of conventional 5X and 10X objectives without repositioning the pedicle screws. To facilitate EC screening *in situ*, we also created 3D printed chamber holders (panel ii, Fig 2A) compatible with most XY microscope stages. These holders prevent the chamber from falling through the stage plate and can be fastened along a stage slide clip for horizontal scanning.

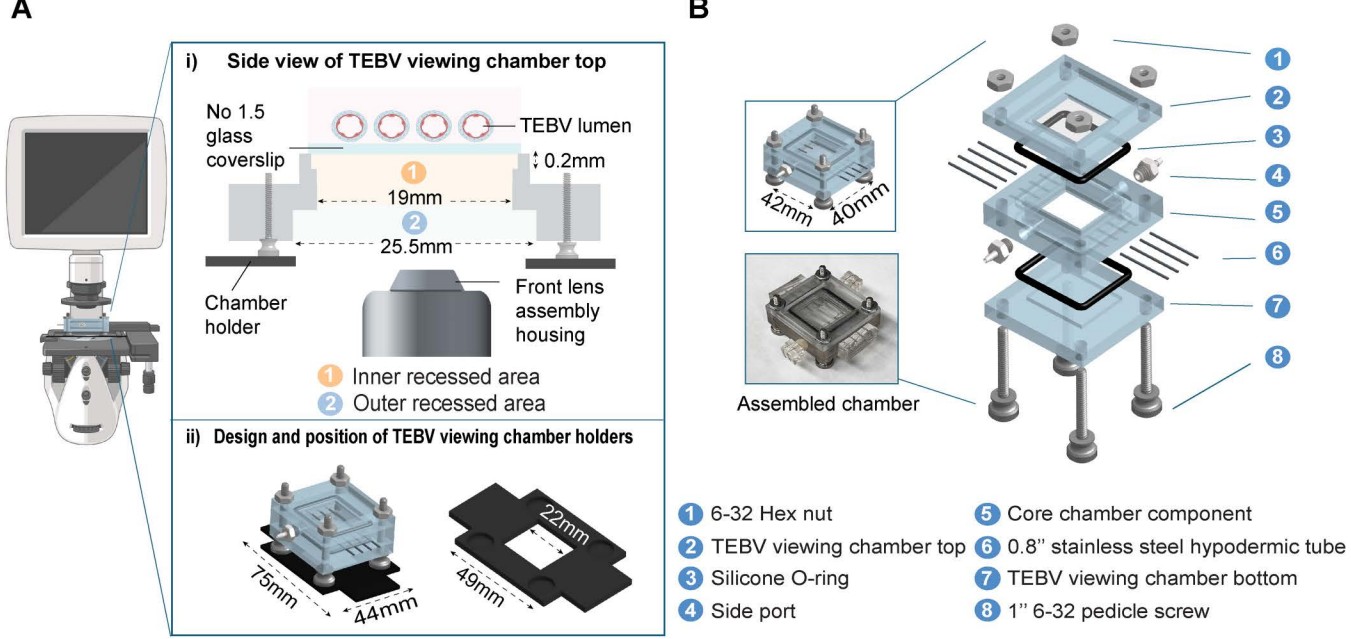

**Fig 2. Design and schematic of the TEBV Chambers and microscope holders.** (A) **i)** Side view of the TEBV viewing chamber top. The outer recessed area is optimized to accommodate the front lens assembly housing, while the inner recessed area is designed to facilitate the observation of 4 TEBVs in the chamber; **ii)** Design and position of TEBV viewing chamber holders. The TEBV chamber is designed to be placed on top of 3D-printed viewing chamber holders. The outer dimensions of the holder are compatible with the slide holder on the microscopy stage, facilitating movement of the chamber during imaging. A central cutout in the holder allows for imaging of all four TEBVs within the chamber. **(B)** Schematic for the TEBV viewing chamber with associated components. The viewing chamber top features a square void designed to accommodate a thin glass slide (part 2, dimensions: 17.5 mm × 19 mm) for direct visualization. The core chamber component (part 5) accommodates four TEBVs. Positioned at the chamber corners are four pedicle screws and nuts (part 1 and 8), while O-rings (part 3) are inserted into the grooves to ensure a secure seal. The two side ports (part 4) are tailored for side loop perfusion. In experiments where imaging capabilities are unnecessary, the viewing chamber top can be substituted with the same piece used for the bottom of the chamber (part 7). Figure created in Biorender (https://BioRender.com).

To assemble the viewing chamber, a 22 × 40 mm glass coverslip (No. 1.5) was cut to 19 × 22 mm and then secured to the inner recessed area with multi-purpose waterproof adhesive. The viewing window top (Part 2, Fig 2B) maintains the same outer dimensions as the perfusion chamber bottom (Part 7, Fig 2B) to permit attachment to the core chamber component (Part 5, Fig 2B). The viewing window panel was only used on one side of the TEBV chamber to minimize potential for media leakage at the site of the glass coverslip, with the original perfusion chamber bottom (Part 7, Fig 2B) on the opposite side. The viewing chamber can be used with both upright and inverted microscopes by simply flipping the position of the window viewing panel. The viewing chamber allows for visualization of all 4 TEBVs, spanning from the left to the right end of the chamber. All viewing chamber designs were generated using the open-sourced design software program, OnShape (www.onshape.com), and can be readily modified to accommodate various microscopes.

## B. Removable PDMS clamps streamline TEBV securing process

The removal of mandrels and subsequent securing of TEBVs within the chamber are a critical step in the workflow process. We designed removable PDMS-based clamps to secure TEBVs within the chamber during perfusion. These clamps obviate the need to secure TEBVs using suturing techniques that are prone to failure. Rectangular molds for PDMS clamps were designed with four equally spaced extruded half-circles (Fig 3A) equivalent to the distance between the four mandrels of the middle chamber component (Part 6, Fig 2B). This design allows for clamps to be positioned above and below mandrels in order to form a tight seal around

the TEBVs (4 clamps per chamber, Fig 3B). Molds for PDMS clamps were printed using a Stratsys J750 3D printer with VeroPure White filament (acrylic formulation). Following PDMS casting, a straight edge razor can be used to removed excess PDMS and to adjust the height (~2.5–3.0 mm) to ensure the clamps fit snugly when the chamber is fully assembled. The clamping force is generated by securely and tightly assembling the chamber. Lateral sides of the clamps can be cut to approximately 1 mm between the edge of each groove and the edge of the clamp. PDMS clamps are reusable and can be steam sterilized.

## C.  Optimized workflow improves EC adherence within TEBV lumens

Injection of a high-density EC suspension is one of the most commonly employed techniques for EC seeding in TEBVs [20,21]. Despite its prevalence, there exists no consensus on the optimal density of ECs required to establish a stable endothelium. In our model system, TEBV endothelialization is accomplished using dynamic rotational seeding [4,19]. The experiments detailed in the following section outline the optimized conditions of dynamic rotational seeding. By using the viewing chamber, endothelialization of TEBVs was monitored during the fabrication step and over the course of 7 days of perfusion. To streamline the imaging process and facilitate rapid screening, we utilized commercially available red fluorescent protein (RFP)-labeled HUVECs from Angio-Proteomie.

**Determination of EC seeding density and rotational seeding duration in TEBVs.**  To determine the optimal conditions for EC attachment prior to applied perfusion, we examined the impact of increasing EC seeding densities and the duration of rotational seeding. Based on our prior work [4,22], we compared three EC seeding densities: low ($1 \times 10^5$ ECs/cm²), medium ($1.5 \times 10^5$ ECs/cm²), and high ($2 \times 10^5$ ECs/cm²). Following luminal injection of the EC suspension, TEBVs underwent either short (3 hours) or long (12 hours) rotation periods at a fixed speed (24 rotations per hour). The percentage of EC luminal coverage within TEBVs was determined by analyzing fluorescence images of fixed, embedded TEBV cross-sections using Fiji software [23]. First, the total luminal length was measured by tracing the entire TEBV lumen perimeter. The extent of EC coverage was then determined by tracing regions with red fluorescence from RFP-HUVECs. The percentage of EC luminal coverage was calculated using the following equation:

$$\frac{\text{Length covered by RFP} - \text{HUVEC}}{\text{Luminal perimeter}} \times 100 = \text{Percent circumferential coverage}$$

A 3-hour rotation period resulted in asymmetric EC coverage within the lumen, regardless of the EC seeding density. Conversely, the 12-hour rotation yielded a significant increase in EC luminal coverage, indicating the need for sufficient

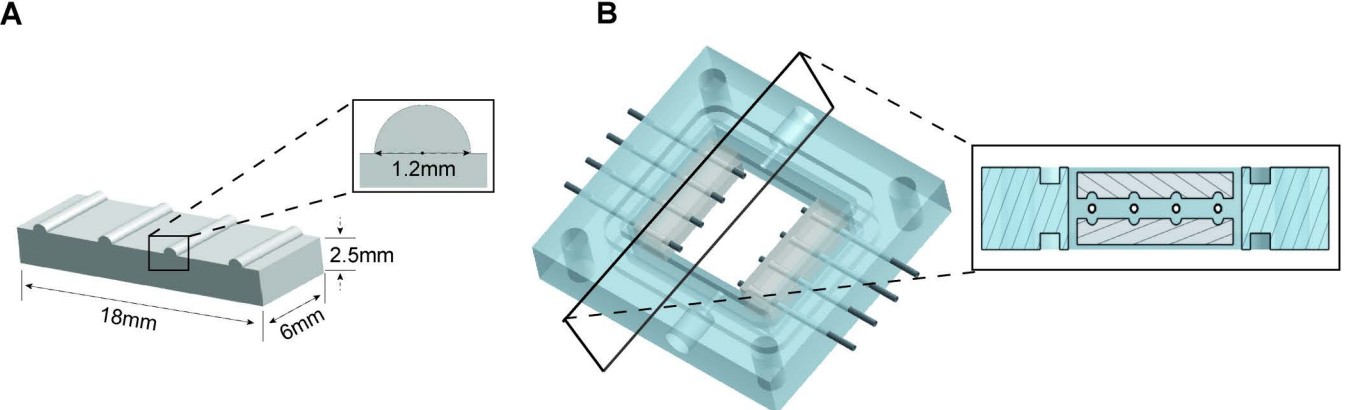

**Fig 3. Design and schematic of the TEBV PDMS clamps. (A)** Mold for polydimethylsiloxane (PDMS) clamps. **(B)** Schematic of the PDMS clamps positioned within the TEBV chamber, showing how four clamps are required to securely hold the TEBVs in place.

rotation to achieve a uniform endothelium in the TEBVs (Fig 4A). Further luminal coverage was significantly higher for a 12 h rotation than for a 3 h rotation, but was independent of seeding densities between 1x10^5 – 2x10^5 cells/cm^2. (Fig 4B). Based on these findings, we elected to explore whether a shorter rotation duration would result in a comparable EC coverage at the same cell seeding density. Following the injection of $1.5 \times 10^5$ ECs/cm^2 per TEBV, TEBVs subjected to only 6 hours of rotation exhibited higher EC coverage in the lumen compared to those with 3-hour rotation periods and

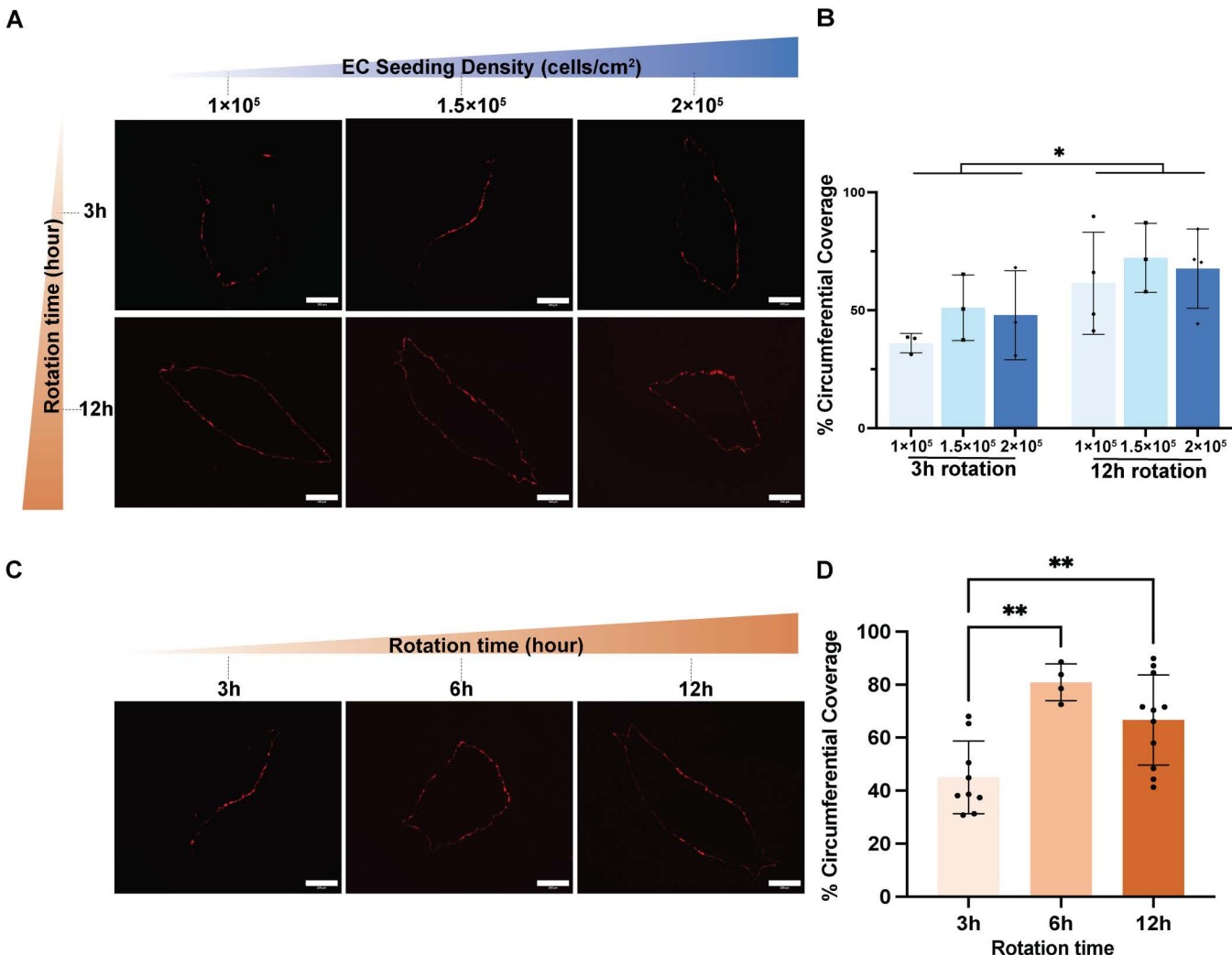

**Fig 4. Increased endothelial cell seeding density and extended rotation duration lead to more uniform endothelial cell coverage within TEBV lumens under static conditions. (A)** Representative images of TEBV cross-sections injected with RFP-HUVECs at low ($1 \times 10^5$ ECs/cm^2), medium ($1.5 \times 10^5$ ECs/cm^2), and high ($2 \times 10^5$ ECs/cm^2) densities, with rotation durations ranging from 3 to 12 hours. Asymmetric adhesion of RFP-HUVECs is observed in the 3-hour rotation group. Scale bar: 200 μm. **(B)** Assessment of circumferential coverage revealed increased luminal coverage in the 12-hour rotation group. TEBVs fabricated with cell density of $1.5 \times 10^5$ ECs/cm^2 with a 12-hour rotation period resulted in the highest endothelial cell coverage. Significance was determined by two-way ANOVA. Mean ± S.D., n = 3 TEBVs (3-hour rotation), n = 4, 3, 4 TEBVs (12-hour rotation). *$P < 0.05$ **(C)** Representative images of TEBV cross-sections injected with $1.5 \times 10^5$ ECs/cm^2 and rotated for 3, 6, 12 hours before fixation. Scale bar: 200 μm. **(D)** Evaluation of circumferential coverage of TEBVs as a function of rotation duration. Prolonged 12-hour perfusion did not lead to a significant increase in EC coverage under static culture. Mean ± **S.**D., n = 9, 4, 11 TEBVs. Since the two-way ANOVA did not detect an effect due to seeding density between 1-2 × 10^5 cells/cm^2, the coverage at different seeding densities in Fig 4B were pooled for the 3-hour and 12-hour rotation groups and compared with the 6 h rotation time at $1.5 \times 10^5$ cells/cm^2 using a one-way ANOVA. **$P < 0.01$. The raw data used to generate this figure are provided in Supporting Information S4 Data.

comparable to coverage with 12 hour rotation under static culture. The 6-hour rotation period also resulted in uniform EC coverage within the TEBV lumen (Fig 4C-4D). Taken together, these findings suggest that for the development of a continuous and functional endothelial layer, a dynamic rotational seeding period (6–12 hours) is needed with a medium cell seeding density.

**Application of increasing shear stress over time for higher endothelial retention within TEBVs.** While previous studies used similar perfusion schemes in TEBVs, no systematic comparison was conducted. Therefore, we here compared two methods for applying perfusion in the TEBV system. Both short term (24 hours) and long term (7 days) effects were analyzed. The first method was a "rapid start" approach, where the pump was set to targeted flow rate of 0.5 ml/min per TEBV (0.4 Pa wall shear stress) immediately after connecting the vessels to the luminal perfusion loop. The equation used to calculate wall shear stress is provided in the protocol (Supporting Information S2 Protocol). The second method involved a "gradual ramping" approach, allowing the flow rate to reach the same target value over a 3-hour period by increasing the flow rate (from 0.04 to 0.4 Pa wall shear stress) in stepwise increments every 60 minutes (Fig 5A).

After fabricating TEBVs using the optimized EC seeding density ($1.5 \times 10^5$ ECs/cm$^2$) and rotation duration (6 hours), we compared the two flow application methods: rapid start and gradual ramping (Fig 5A). To evaluate the immediate effect of perfusion, TEBVs were then subjected to 24 hours of perfusion (0.4 Pa wall shear stress/TEBV) and the extent of EC coverage was compared. After 6 hours of rotation, the rapid start perfusion group showed less EC coverage compared to the gradual ramping perfusion group, suggesting that ECs require more time to adhere to the collagen substrate before being exposed to shear stress. A similar trend was also observed in the 12-hour rotation group. In contrast, with gradual ramping perfusion, TEBVs subjected to 12-hour rotation achieved luminal EC coverage achieved slightly higher coverage than after a 6-hour rotation, although the difference was not significant (Fig 5B). We postulated that a 6-hour rotation may not be sufficient for ECs to fully anchor to the underlying collagen substrate, making them less resilient to the applied shear stress.

Thus, 7-day perfusion experiments were performed using the medium EC seeding density ($1.5 \times 10^5$ ECs/cm$^2$), 12-hour rotation duration, and gradual ramping of the flow rate. Compared to TEBVs subjected to rapid initiation of perfusion, TEBVs exposed to a gradual increase in perfusion also exhibited greater circumferential EC coverage after 7 days (Fig 5C-5D). These findings indicate that those ECs that have not fully adhered to the collagen substrate or established stable cell-cell junctions are susceptible to detachment when subjected to a rapid onset of perfusion. In contrast, a gradual increase in perfusion allows ECs sufficient time to sense and adapt to the escalating shear stress. This progressive adaptation improves cell adhesion and retention on the vessel wall, supporting the development of a stable and uniform endothelial monolayer.

**Theoretical evaluation of rotational speed during endothelial cell seeding.** With the rotational speed fixed by the motor, we ran simulations to explore how rotation speed and cell density affect cell distribution on the inner surface of TEBVs. Previous studies have analyzed the behavior of cells and beads in a rotating bioreactor to simulate microgravity conditions [24]. These studies focused on determining the appropriate rotational speed to keep microparticles or beads suspended, counteracting the settling force due to the density difference between the particle (or cell) and the surrounding fluid. In our case, the objective is to find the rotational speed that ensures uniform cell deposition on the lumen of the TEBVs. We present the simulation results for a range of rotational speeds which may also guide others in designing a rotator for similar systems.

Assuming that the fluid flow field is not affected by the cells, the equations of motion for the cell are as follows.

Radial direction $M \left( \frac{dv_r}{dt} - r \left( \frac{d\theta}{dt} \right)^2 \right) = -m \left( C_v + 1 \right) r\omega^2 + (m - m_p) \, g sin\theta - 6\pi\mu a v_r$

Transverse $(\theta)$ direction $M \left( r \frac{d^2\theta}{dt^2} - 2v_r \frac{d\theta}{dt} \right) = -(m - m_p) \, cos\theta - 6\pi\mu a r \left( \frac{d\theta}{dt} - \omega \right)$

Where $M = m_p + C_v m = r_p V_p + C_v r_p V_p$, $m_p$, $r_p$ and $V_p$ are the total mass, density, and volume of the cell (or particle), $C_v$ is the virtual mass coefficient (0.5 for a sphere) and $a$ is the cell radius.

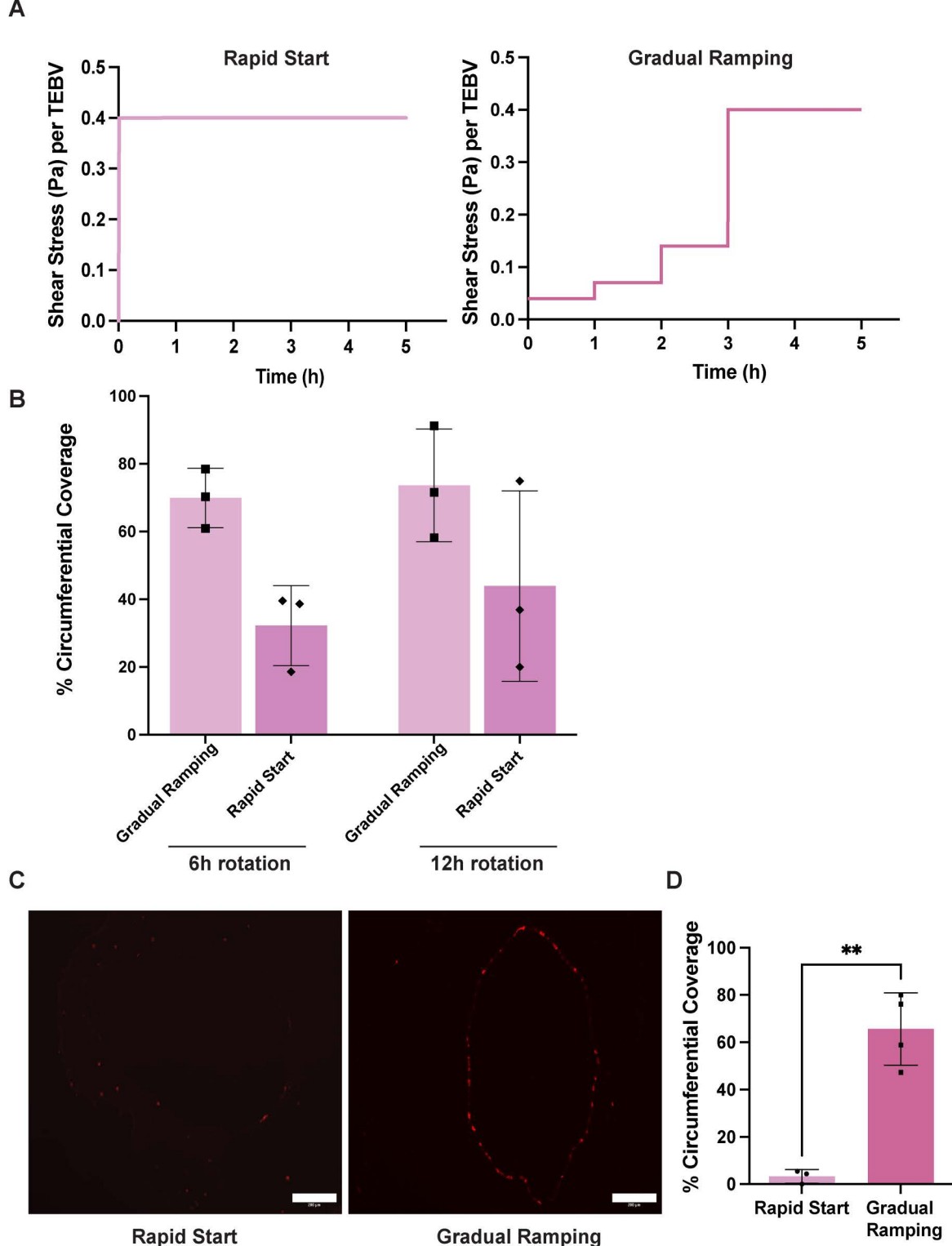

**Fig 5. Gradual ramping perfusion promotes endothelium stability and rapid start perfusion causes endothelial cell loss. (A)** Shear stress applied to each TEBV over a 3-hour period of flow application. The graph on the left shows a rapid increase to a steady shear stress, while the graph on the right illustrates a stepwise increase in shear stress. **(B)** Evaluation of circumferential EC coverage between TEBVs with different rotation periods and

flow application methods. TEBVs in the 24h perfusion groups were subjected to the two different perfusion schemes after EC seeding. n=3 TEBVs. Data were analyzed using a two-way ANOVA, and no significant differences were observed. **(C)** Representative images of TEBV cross-sections injected with RFP-HUVECs at the density of $1.5 \times 10^5$ ECs/cm² and rotated for 12 hours. After 7 days perfusion at a flow rate of 0.5 ml/min per TEBV (0.4 Pa shear stress), no EC signal was observed in the rapid start group. Scale bar: 200 μm. **(D)** TEBVs subjected to a gradual increase in perfusion exhibit greater circumferential EC coverage when compared to the group with rapid initiation of perfusion after 7 days. Significance was determined by Student's t-test. mean±S.D., n=3, 4 TEBVs. **P<0.01. The raw data used to generate this figure are provided in Supporting Information S4 Data.

The virtual mass arises from the inertia on a particle due to the surrounding fluid. Some fluid moves along with the particle adding to the total particle mass. The term $m - m_p$ can be expressed as $\Delta\rho V_p$, where $\Delta\rho$ is the difference in density between the cell and culture media, approximately 0.05 g/cm³. Given the rotation speed that we used for TEBV endothelialization (24 rph) and the inner radius of the TEBVs (~0.04 cm), the flow is laminar ($Re = 0.0158$). Thus, the overall fluid motion is one of solid body rotation, described by $v_\theta = r\omega$. Since the cell density is greater than the fluid density, there is a tendency for the cell motion to deviate from the fluid motion at the same location.

To validate this model, we replicated the conditions reported for microcarriers in a rotating wall bioreactor [25]. The results shown in Fig 6 closely matched the reference, where a particle denser than the surrounding fluid slowly rotates towards the wall with each rotation.

We then analyzed the cell trajectory under several cases, where the cell radius was 7.5 μm, the inner TEBV radius was 0.04 cm, and the density difference was 0.05 g/cm³. For a 1 cm-long TEBV, the inner luminal surface area is 0.25 cm² and the lumen volume is 0.005 cm³. First, we varied the rotational speed and examined the trajectory of a cell at radial position of 0.001 cm, 0.005 cm, 0.01 cm or 0.02 cm starting at $\theta = 90°$. At 15 rph, the cells at each initial position contact the TEBV inner radius within one revolution (Fig 7A). At 20 rph, cells at the same initial positions behave similarly (Fig 7B). Interestingly, unlike the larger microparticle, if the cell does not contact the inner surface of the TEBV in its first revolution, it will continue to follow the same trajectory in subsequent rotations. At a rotation speed of 24 rph, only cells starting at a radial position of 0.02 cm come into contact with the TEBV wall (Fig 7C). This suggests that the likelihood of a cell contacting the vessel wall depends on its initial radial position, which may explain why longer durations are required to achieve uniform coverage. Speeds higher than 29 rph prevent cells from contacting the vessel wall, setting an upper limit on the rotational speed (Fig 7D). These results indicate that the rotational speed we used for endothelization are sufficient for ECs to contact and attach to the inner lumen.

We further examined how varying cell density affects the trajectory, using a density of 1.05 g/cm³ (blue) and 1.2 g/cm³ (red), both initially positioned at $r = 0.005$ cm. At 15 rph, the denser cell followed a larger arc in its movement (Fig 7E). Higher rotational speeds produced a more confined trajectory pattern, making them less likely to come into contact with the vessel wall (Fig 7F). This finding implies that while cell density influences trajectory, rotational speed is the main factor in influencing EC distribution. Lower rotational speeds facilitate EC contact with the vessel wall. Thus, the rotational speed must be carefully adjusted to ensure uniform cell distribution within the vessel.

To determine optimal EC seeding conditions, we not only analyzed pre-adhesion cell trajectories in the rotating TEBVs but also examined the forces acting on newly adherent ECs and evaluated their potential to induce cell detachment, providing further insights into the rotational speeds used. There are two primary forces acting on the ECs within TEBVs: the drag force exerted by the rotating fluid on a spherical cell and the gravitational force. The drag force can be expressed as $F_{Drag} = 6\pi\mu ar\omega$, where $\mu = 0.0085$ g/(cm·s) is the media viscosity at 37°C, $a = 7.5$ μm is the cell radius, $r = 0.04$ cm is the TEBV luminal radius, and $\omega$ is the angular velocity corresponding to 24 rph. For these conditions, $F_{Drag}$ is calculated to be approximately $2.01 \times 10^{-12}$ N. The gravitational force acting on an adherent EC varies with the angle in the TEBV lumen and is expressed as $F_{Gravity} = m_p g \cos(\frac{\theta}{2})$, where $m_p$ is the cell mass, $g = 9.8$ m/s², and $\theta$ corresponds to the cell's position. The maximum gravitational force occurs at the top of the lumen where $\theta = 0°$. For a cell with radius $a = 7.5$ μm and density $\rho = 1.05$ g/cm³, the maximum gravitational force is $1.82 \times 10^{-11}$ N. Previous studies suggested that a shear

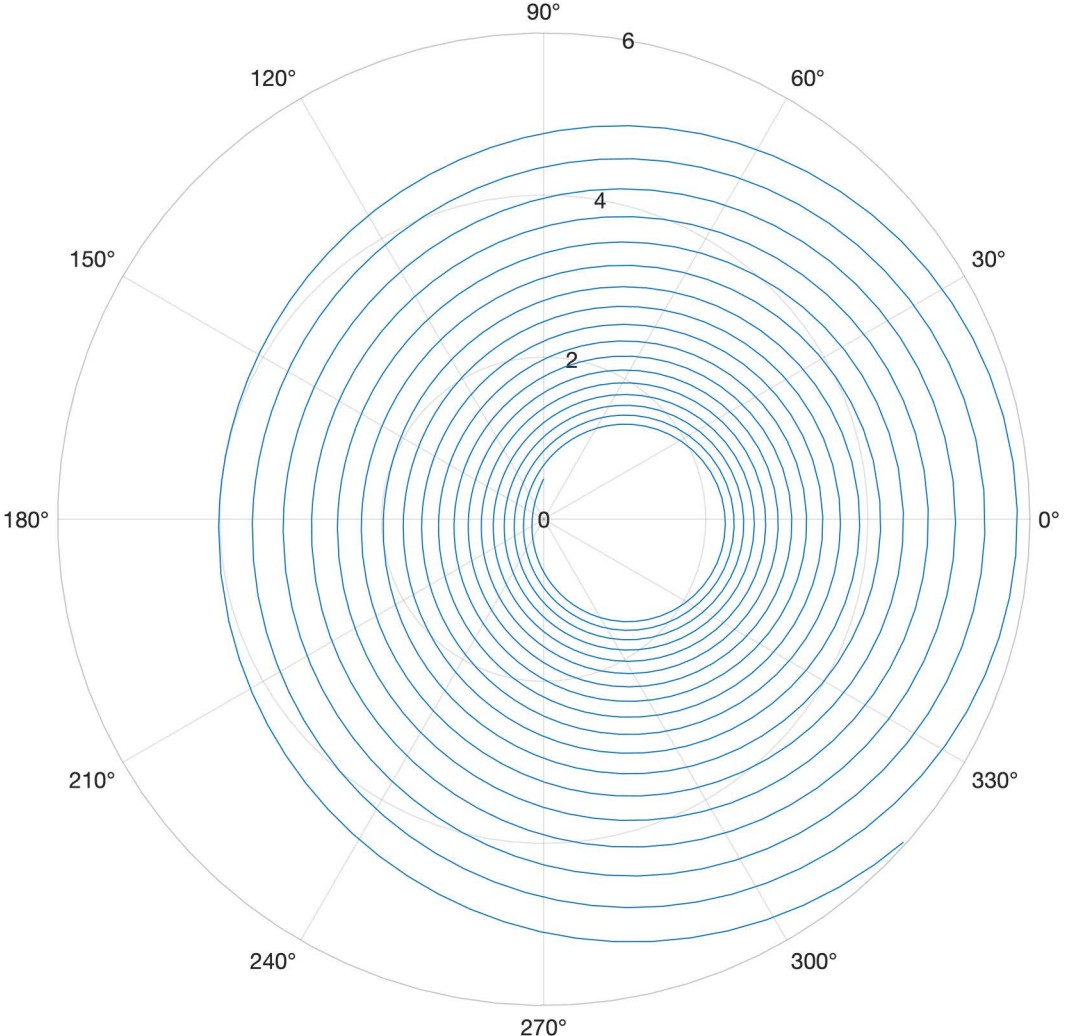

**Fig 6. The trajectory of a microcarrier bead under specific rotational conditions.** Parameters included a rotational speed of 35 rpm, fluid viscosity $\mu = 0.01$ g/(cm·s), microcarrier radius $a = 330$ µm, and density of the microcarrier, $\rho_p = 1.3$ g/cm³. The rotating wall vessel radius was 5 cm. The bead started at an initial position of r = 0.5 cm and 90°.

stress of 0.01 Pa generates a force of $1.57 \times 10^{-11}$ N while a shear stress of 0.1 Pa equals $1.57 \times 10^{-10}$ N [26]. These levels of shear stress are sufficient to induce detachment of newly adherent cells. Based on these calculations, the gravitational force may destabilize cells located at the highest point of the lumen, particularly when the adhesion strength is low. Using a rotational speed less than or equal to 29 rph ensures that the drag force from fluid rotation remains below the threshold required to detach adherent cells.

### D. TEBVs generated using this protocol were found to support a functional endothelium

A comprehensive evaluation was conducted by monitoring luminal EC coverage for 7 days. There were no significant changes in EC coverage within TEBVs at 24 hours, 48 hours, 72 hours and 7 days based on cross-section evaluation, revealing that our protocol supports the development of a stable endothelium under physiologically relevant perfusion (Fig 8A). By using the viewing chamber, we also tracked the endothelialization of TEBVs over 7 days of perfusion and

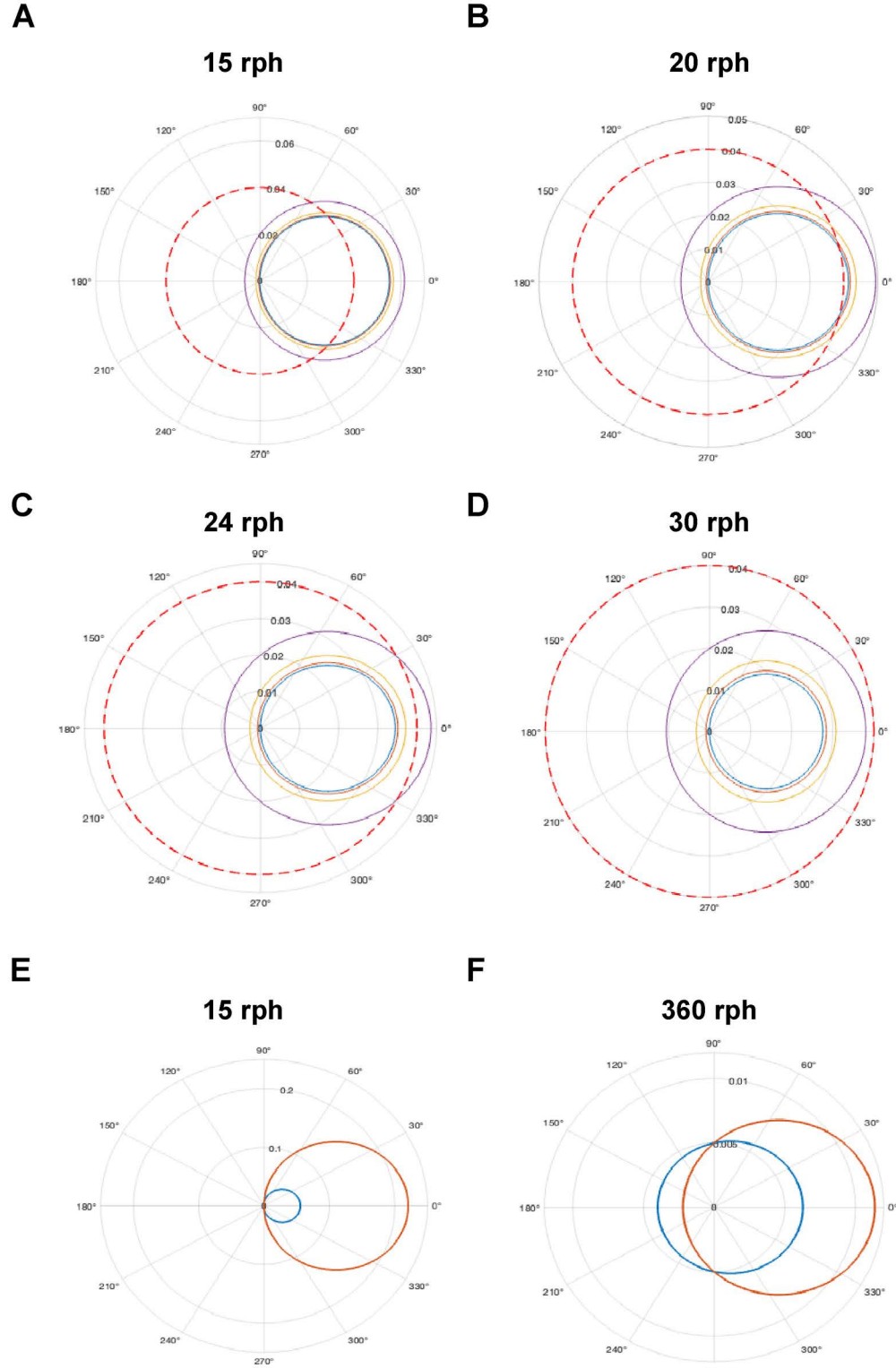

**Fig 7. Effect of initial radial position and particle density on movement of a particle in a TEBV. (A)** The trajectories of cells initiated from radial positions of 0.001 cm, 0.005 cm, 0.01 cm, and 0.02 cm at a rotation speed of 15 rph. The viscosity of cell culture medium at 37°C was 0.0085 g/(cm·s). The inner TEBV wall, located at 0.04 cm, is indicated by a dashed red line. **(B)** The trajectories of cells starting from the same radial positions of

0.001 cm, 0.005 cm, 0.01 cm, and 0.02 cm at an increased rotation speed of 20 rph. The viscosity of cell culture medium at 37°C was 0.0085 g/(cm·s). The inner TEBV wall is highlighted by a dashed red line. **(C)** The trajectories of cells initiated from radial positions of 0.001 cm, 0.005 cm, 0.01 cm, and 0.02 cm at a rotation speed of 24 rph. The inner TEBV wall, located at 0.04 cm, is indicated by a dashed red line. **(D)** At 30 rph, the trajectories of cells starting from the same radial positions of 0.001 cm, 0.005 cm, 0.01 cm, and 0.02 cm. The dashed red line marks the inner TEBV wall at a radius of 0.04 cm. **(E)** At 15 rph, the trajectory represented in red corresponds to cells with a higher density, which forms a larger circular pattern compared to the trajectory of cells with a density of 1.05 g/cm³. **(F)** At a high rotation speed of 360 rph, the cell trajectories become more confined, indicating that the rotation speed has an impact on the trajectory formation, irrespective of the cell density.

fluorescent lumens were observed (Fig 8B). Immunostaining for von Willebrand factor (vWF) in TEBV cross-sections further confirmed the presence of an intact endothelial layer (Fig 9A). *En face* staining of TEBVs also demonstrated prominent VE-cadherin at cell-cell junctions, indicating that ECs in TEBVs respond to hemodynamic forces and form tight junctions (Fig 9B). The EC alignment in the direction of flow also validated the responsive nature of ECs within the TEBV lumen (Fig 9C).

Understanding the inflammatory response is pivotal for assessing blood vessel function. Hence, we investigated the effect of TNF-α, a potent inflammatory stimulus, on EC activation in TEBVs. We observed a significant increase in THP-1 monocyte accumulation within the TEBV lumen following 200U/mL TNF-α treatment, indicative of EC activation (Fig 10A). Upregulation of Intercellular Adhesion Molecule-1 (ICAM-1), a mediator of monocyte-EC interaction, was also appreciated compared to untreated TEBVs as determined by immunostaining (Fig 10B). Collectively, these findings indicate that ECs within the TEBVs are functionally responsive to inflammatory stimuli similar to native vessels, highlighting the promising functionality of the engineered endothelium.

## Discussion

Microphysiological platforms have emerged as promising models to enhance our understanding of vascular disease pathology and facilitate drug screening [27]. Among these, TEBVs using human-derived cells have shown to be effective model systems for a variety of disease states [22,28]. Despite notable advancements in vascular engineering, there

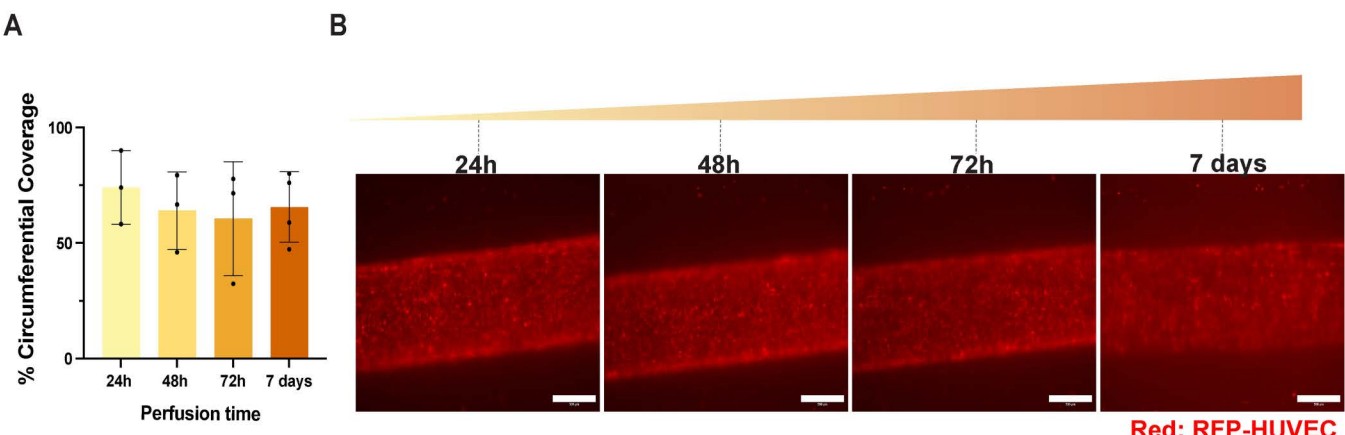

**Fig 8. Evaluation of endothelialization and *in situ* imaging of TEBVs over time. (A)** Evaluation of endothelialization at selected timepoints. RFP-HUVECs remained adhered throughout the duration of perfusion. mean ± S.D., n = 3, 3, 3, 4 TEBVs. Data were analyzed using a one-way ANOVA, and no significant differences were observed. The 7-day gradual ramping results presented here are the same as those shown in Fig 5D, as these experiments were conducted simultaneously. **(B)** *In situ* imaging of TEBVs over a 7-day perfusion period using the 5X objective lens. Scale bar: 500 μm. The raw data used to generate this figure are provided in Supporting Information S4 Data.

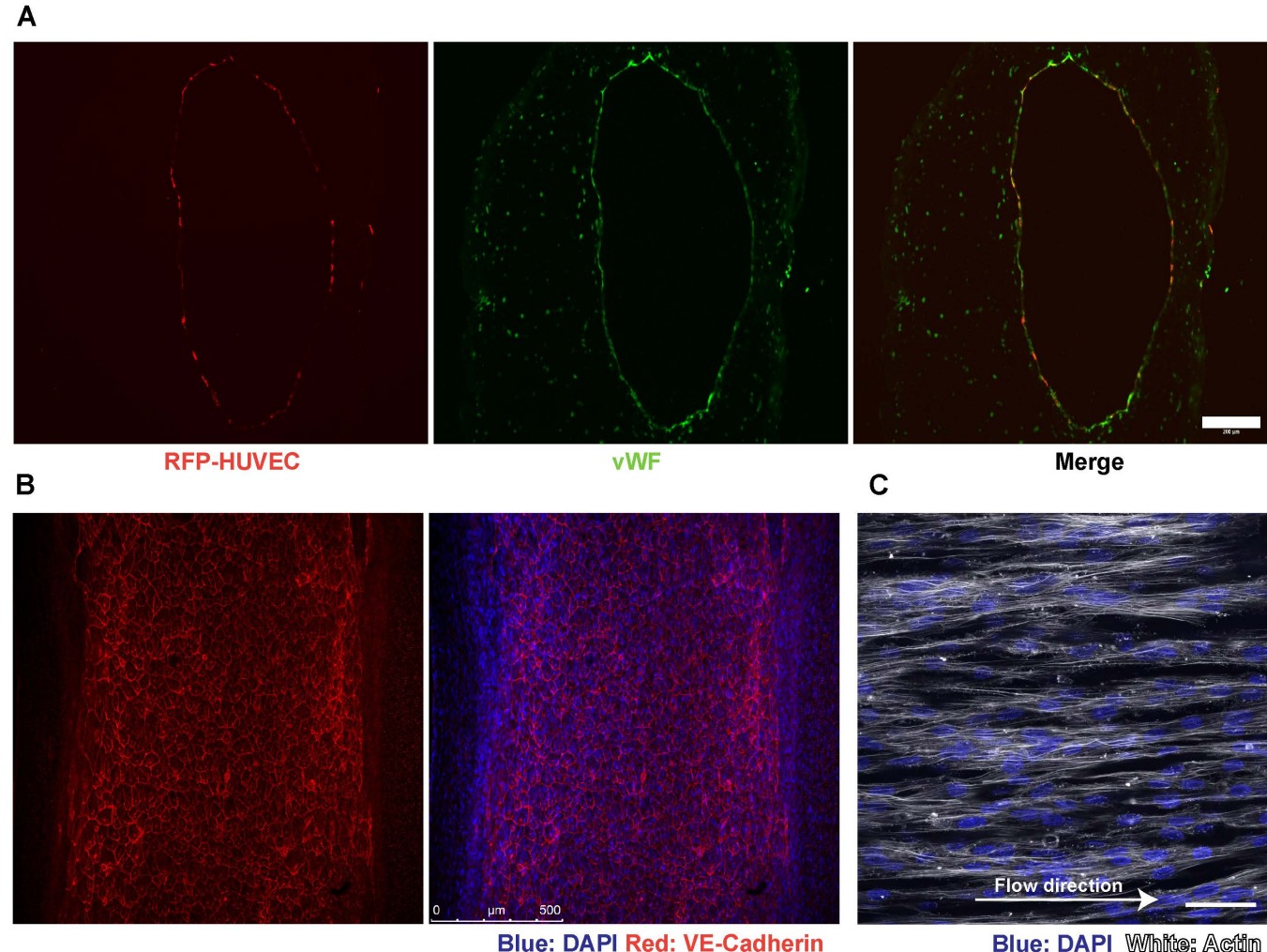

**Fig 9. Maintenance of endothelium within TEBVs following perfusion. (A)** Cross-sectional view of TEBVs fabricated with $1.5 \times 10^5$ ECs/cm$^2$ and perfused at a flow rate of 0.5 ml/min per TEBV (0.4 Pa shear stress) for 7 days. TEBVs show intact endothelium positive for vWF by immunostaining. Scale bar: 200 μm. **(B)** *En face* sections of TEBVs show HUVECs forming tight junctions with VE-Cadherin localization and aligning with the flow direction in the lumen after 7 days of perfusion under 0.4 Pa shear stress. Scale bar: 200 μm, 50 μm. **(C)** ECs show alignment in the direction of flow after perfusion.

remains a significant gap in accessible chamber designs and experiment procedures for fabricating and maintaining engineered vessels. In this study, we aimed to address these challenges by applying a novel viewing chamber that facilitated the monitoring of endothelialization conditions of TEBVs *in situ*.

Achieving a stable endothelium is a prerequisite for mimicking vascular physiology. In this study, we refined the fabrication and perfusion protocol of TEBVs with a focus on improving endothelialization. We demonstrated that the inclusion of a 12-hour rotation period and a gradual application of flow were pivotal factors in enhancing EC retention following the initiation of shear stress. This approach resulted in the formation of a robust endothelium and facilitated the alignment of ECs in the flow direction. Our rotational seeding duration corresponds to previous findings that an increased duration results in an increased seeding efficiency with an optimal duration of 12 hours [17]. Previous studies have also indicated that a gradual application of flow after an initial exposure to low shear stress prevents disrupted endothelial coverage observed after immediate exposure to physiological levels of shear stress [29,30]. Our model simulations and calculations

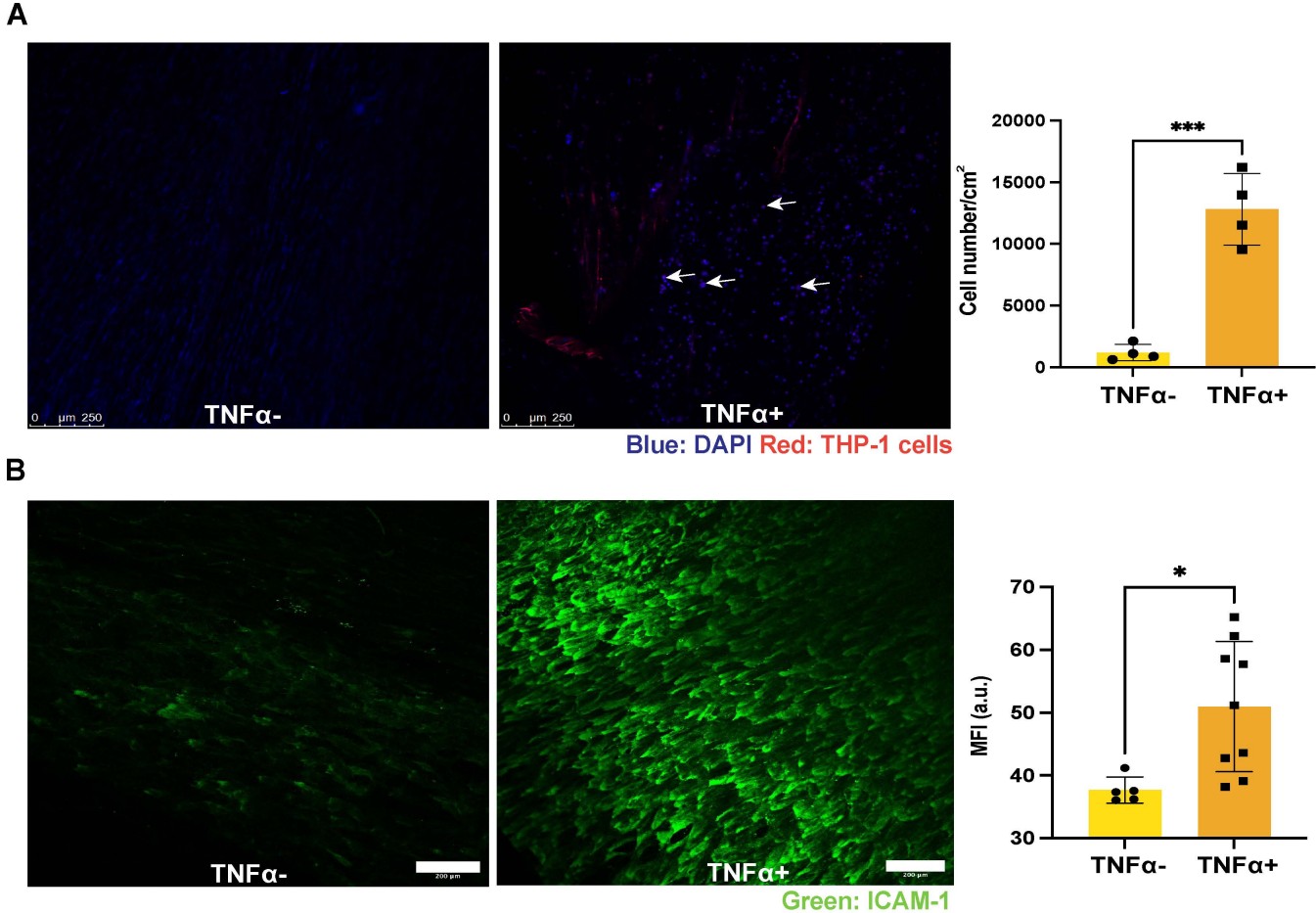

**Fig 10. Endothelial cells in the TEBVs respond to inflammatory stimulus. (A)** THP-1 monocytes (white arrows) show increased adhesion to endothelium after TNF-α treatment. Scale bar: 250 μm. Significance was determined by Student's t-test. mean±S.D., n=4 TEBVs. *P<0.05 **(B)** Quantification of ICAM-1 expression in TEBVs after TNF-α treatment. Scale bar: 200 μm. mean±S.D., n=5, 9 TEBVs. ***P<0.001. The raw data used to generate this figure are provided in Supporting Information S4 Data.

evaluating cell trajectories and adhesion under different conditions provided justification for the rotational speeds. We demonstrated that the selected rotational speeds are sufficient for effective cell adhesion and distribution. These results provide a framework for determining optimal bioreactor parameters to achieve uniform cell distribution in tissue-engineered vessel constructs. The microscopic analysis confirmed the presence of vWF positive endothelium with VE-cadherin staining indicating successful cell-cell junction formation, further substantiating the successful endothelialization of TEBVs under the optimized protocol. The 7-day maturation period under flow condition resulted in TEBVs with physiological relevance, as evidenced by their responses to inflammatory signals. Endothelium activation with TNF-α treatment led to increased monocyte adhesion. In line with previous studies [31], ECs upregulated ICAM-1 expression when exposed to the inflammatory stimuli. This aspect of our work underscores the clinical significance of the developed TEBVs, showcasing their potential to mimic the dynamic interactions between blood vessels and the immune system.

A critical feature for tissue engineered systems is the ability to conduct real-time observation, which is often hindered by the design of the chamber, particularly due to the large volumes and thicknesses that exceed the working distance of most microscopy objectives [32,33]. The novel viewing chamber developed in our study has significantly enhanced

the efficiency of the TEBV system. First, by enabling *in situ* screening during the endothelialization process, this design reduces the error rate and circumvents unnecessary downstream procedures, thereby conserving valuable biological samples, and other costly reagents. The viewing chamber pieces are also compatible with the perfusion chamber, avoiding the use of excessive volume of medium required for system perfusion and enabling efficient imaging of up to 4 TEBVs simultaneously, compared to the previous design [34].

Furthermore, the current design offers the potential for observing cell dynamics by incorporating flow-induced shear and spatial cell arrangements that closely mimic *in vivo* conditions. Previous studies have either modified existing chambers or designed new perfusion chambers to enable visualization within a single TEBV [34,35]. Our design has an advantage over these previous perfusion chambers by enabling the visualization of four TEBVs within a single perfusion chamber. The existence of multiple replicates within a chamber and our screening process improves throughput and reproducibility coupled with a reduction of required resources. By providing designs for the viewing chamber and chamber holders using open-sourced software, we outline a customizable solution that can be tailored to specific research needs.

Despite these advancements, this study has several limitations. First, we did not re-evaluate the viability of the system for perfusion durations exceeding 7 days. In prior studies, we demonstrated that TEBVs can be maintained in culture for up to 4 weeks and assessed for endothelial function, showing that these vessels can promote vasodilation in response to acetylcholine administration [16,22]. Secondly, although the use of PDMS clamps and viewing chamber improves the ability to secure TEBVs conveniently and allow for *in situ* observation, the depth of observation might still be limited by the thickness and opacity of the TEBVs as shown in Fig 8B. This could restrict detailed imaging of deeper layers or regions of the vessel, which might be crucial for understanding certain pathological processes or cellular interactions. Future work will focus on optimizing the viewing chamber design to enhance compatibility with confocal microscopy and a wider range of objectives. Lastly, while the rotation speed has been validated through simulation, we did not investigate its impact on the endothelialization outcome due to the lack of a rotator with adjustable speed settings. However, previous studies have suggested that higher rotation velocities may inhibit cell attachment, while lower speeds can lead to uneven cell distribution [21]. Simulations and experiments will be necessary to determine the optimal combinations for varying parameters in other systems. For example, our simulation results suggest that vessels with larger diameters may require slower rotation speeds for cells to contact the vessel wall.

In summary, the TEBVs described in this study have proven to be valuable and convenient models for recapitulating human blood vessels. We anticipate that the protocol outlined here will further optimize tissue-engineered vascular models, enhancing their ability to simulate the complex dynamics and interactions of human physiology, potentially leading to advancements in vascular medicine and regenerative therapies.

## Supporting information

**S1 File. Chamber Drawings. CAD files for generating TEBV chambers, 3D printed chambers holder and PDMS clamp mold.**
(ZIP)

**S2 File. Protocol. Step-by-step protocol for tissue engineering blood vessel fabrication and maintenance.** Also available on protocols.io.
(DOCX)

**S3 Video. Video showing the protocol workflow.**
(MP4)

**S4 Data. Raw data.**
(XLSX)

## Author contributions

**Conceptualization:** Jingyi Zhu, Kevin L. Shores, George A. Truskey, Stacey A. Maskarinec.

**Data curation:** Jingyi Zhu, Halie L. Hotchkiss, Stacey A. Maskarinec.

**Formal analysis:** Jingyi Zhu, George A. Truskey.

**Funding acquisition:** George A. Truskey, Stacey A. Maskarinec.

**Investigation:** Jingyi Zhu, Halie L. Hotchkiss, Kevin L. Shores, George A. Truskey, Stacey A. Maskarinec.

**Methodology:** Jingyi Zhu, Halie L. Hotchkiss, Kevin L. Shores, George A. Truskey, Stacey A. Maskarinec.

**Project administration:** Jingyi Zhu, Halie L. Hotchkiss, George A. Truskey, Stacey A. Maskarinec.

**Resources:** George A. Truskey, Stacey A. Maskarinec.

**Software:** Jingyi Zhu, Halie L. Hotchkiss, George A. Truskey.

**Supervision:** George A. Truskey, Stacey A. Maskarinec.

**Validation:** Jingyi Zhu, Halie L. Hotchkiss, Stacey A. Maskarinec.

**Visualization:** Jingyi Zhu, Halie L. Hotchkiss, George A. Truskey, Stacey A. Maskarinec.

**Writing – original draft:** Jingyi Zhu, Stacey A. Maskarinec.

**Writing – review & editing:** Jingyi Zhu, Halie L. Hotchkiss, Kevin L. Shores, George A. Truskey, Stacey A. Maskarinec.

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
