## [Decision Letter · Decision Letter 0]

17 Jan 2025

PONE-D-24-55852Strategies for improved endothelial cell adhesion in microphysiological vascular model systemsPLOS ONE

Dear Dr. Truskey,

Thank you for submitting your manuscript to PLOS ONE. After careful consideration, we feel that it has merit but does not fully meet PLOS ONE’s publication criteria as it currently stands. Therefore, we invite you to submit a revised version of the manuscript that addresses the points raised during the review process.

We look forward to receiving your revised manuscript.

Kind regards,

Ahmed El-Fiqi, Ph.D.

Academic Editor

PLOS ONE

2. We note that your Data Availability Statement is currently as follows: [All relevant data are within the manuscript and its Supporting Information files.] Please confirm at this time whether or not your submission contains all raw data required to replicate the results of your study. Authors must share the “minimal data set” for their submission. PLOS defines the minimal data set to consist of the data required to replicate all study findings reported in the article, as well as related metadata and methods (https://journals.plos.org/plosone/s/data-availability#loc-minimal-data-set-definition).

Reviewers' comments:

Reviewer's Responses to Questions

**Comments to the Author**

1. Does the manuscript report a protocol which is of utility to the research community and adds value to the published literature?

Reviewer #1: Yes

Reviewer #2: Yes

2. Has the protocol been described in sufficient detail?

To answer this question, please click the link to protocols.io in the Materials and Methods section of the manuscript (if a link has been provided) or consult the step-by-step protocol in the Supporting Information files.

The step-by-step protocol should contain sufficient detail for another researcher to be able to reproduce all experiments and analyses.

Reviewer #1: Yes

Reviewer #2: Yes

3. Does the protocol describe a validated method?

Reviewer #1: Yes

Reviewer #2: Yes

4. If the manuscript contains new data, have the authors made this data fully available?

Reviewer #1: Yes

Reviewer #2: Yes

**5. Is the article presented in an intelligible fashion and written in standard English?**

Reviewer #1: Yes

Reviewer #2: Yes

6. Review Comments to the Author

Reviewer #1: The authors present a lab protocol for optimized endothelial cell seeding within microphysiological vascular model systems (vessel diameters ~0.8 mm). They explore three parameters: (1) endothelial cell seeding density, (2) rotation duration, and (3) method of flow initiation. The parameters they optimize are uniformity and stability of seeded cells upon exposure to 24 hours of fluid shear stress. While previous groups have published protocols for uniformly seeding vascular grafts, the current protocol presents a systematic effort to identify optimal seeding conditions, particularly for microvessels. Furthermore, the protocol presents a flow chamber with innovative design features including a microscopy viewing window and simple vessel clamps. Overall, the manuscript is well written and the experiments are described logically. This is a valuable contribution to the field, especially for groups working with micropysiological vascular model systems. Applicability to the broader field of vascular tissue engineering, which typically utilizes larger diameter vessels, different endothelial cell types, different substrate biomaterials, different adhesion coatings, and different flow conditions, is less certain. Specific comments are below.

1. The rotation speed of the system is described as 15-22 rph and 18-22 rph in different parts of the manuscript. Which is correct and why is there a range?

2. Do the authors expect their optimal conditions to also apply to larger diameter vessels (e.g. 3-6 mm diameter), different endothelial cell types, different biomaterial substrates, different adhesion coatings, and different flow conditions? Might it be necessary to determine optimal seeding conditions for each combination of these parameters?

3. Given that this is a lab protocol manuscript, the detailed step-by-step protocol provided in the supplement is appreciated. However, this is likely too much detail for the typical reader. I suggest providing a high-level overview of the protocol in the Methods section of the manuscript so readers have sufficient information to understand the rest of the manuscript without needing to read the lengthy supplement.

4. The viewing chamber described in Expected Results section A is difficult to understand based on a text description. Please provide a schematic figure with dimensions and labels. What is the distance between the TEBV lumen and the microscope objective? What is the “frontal housing lens?” Example images using a light microscope would be informative.

5. Please provide CAD files for the chamber, chamber holder, and mold designs in the supplement. This will allow others to fabricate similar devices.

6. In Expected Results section B, it would help to clarify the clamping force is provided by snugly assembling the chamber.

7. Please also express the cell seeding densities as cells per cm2 of seeding area. Given the cells are settling onto a surface, the number per area is more useful than the number per volume.

8. Why was en face imaging not performed to calculate cell coverage within the vessels? Why was sectioning performed and circumferential coverage calculated when the chambers nicely provide the ability to directly visualize the cell coverage on the surface of the vessels (e.g. Figure 8B)?

9. Why is there no static group for the data presented in Figure 5C-D?

10. Line 346 may have a typo: starting position in Figure 7A appears to be 90 deg.

11. It is stated that rotational speeds above 29 rph prevent cells from contacting the vessel wall. Can these data be shown? E.g. 30 rph?

12. The last section in Expected Results section C describing the forces acting on the cells reads as disconnected from the rest of the manuscript. Please relate this section back to optimal seedings conditions, especially rotational speed.

13. Please define the white arrows in the caption for Figure 10A.

14. In Figure 1A, consider showing hours/minutes rather than days on the timeline.

15. In Figure 7, consider highlighting the inner wall located at 0.04 cm.

Reviewer #2: Reviewing decision for Manuscript Number: PONE-D-24-55852

This study offers an efficient protocol in tissue-engineered blood vessel modeling through the ability to monitor multiple TEBVs simultaneously, developing a viewing chamber for real-time visualization during endothelialization process, simulating the physiological conditions by gradually ramping shear stress, leading to a more stable endothelial layer, demonstrating an optimized process for seeding density and rotation duration, addressing previous gaps in EC adhesion studies and incorporating functional markers (e.g., VE-cadherin, ICAM-1) to confirm endothelial stability and functionality.

All these provide an accessible workflow for vascular research.

So, I recommend this paper for publication in PLOS One journal.

7. PLOS authors have the option to publish the peer review history of their article (what does this mean? ). If published, this will include your full peer review and any attached files.

**Do you want your identity to be public for this peer review?** For information about this choice, including consent withdrawal, please see our Privacy Policy .

Reviewer #1: No

Reviewer #2: No

---

## [Author Response · Author response to Decision Letter 1]

26 Feb 2025

Dear Reviewers,

We are pleased to submit our revised manuscript, “Strategies for improved endothelial cell adhesion in microphysiological vascular model systems (PONE-D-24-55852)” for consideration for publication in PLOS One Lab Protocols. We thank the reviewers for providing helpful feedback and insightful comments. Please find below our point-by-point response to the reviewers. All line numbers referenced correspond to the tracked-changes version of the manuscript.

Reviewer #1

1. The rotation speed of the system is described as 15-22 rph and 18-22 rph in different parts of the manuscript. Which is correct and why is there a range?

RESPONSE: We thank the reviewer for the comment. The correct rotation speed range is 18-22 rph. This range reflects the operation of the two custom-made rotators used in our study. Based on our measurements, the average time taken for one rotation is approximately 2.5 minutes, which corresponds to a rotation speed of around 24 rph under ideal conditions. We have included the simulation result for 24 rph in Fig 7C. However, slight fluctuations in speed are expected, and the reported range serves as a practical reference for others who may wish to build similar rotators. We have revised the manuscript to ensure consistency and provide this clarification (lines 115-119).

2. Do the authors expect their optimal conditions to also apply to larger diameter vessels (e.g. 3-6 mm diameter), different endothelial cell types, different biomaterial substrates, different adhesion coatings, and different flow conditions? Might it be necessary to determine optimal seeding conditions for each combination of these parameters?

RESPONSE: While the presented conditions were optimized for the current system, we believe the general conclusions are applicable to other systems. However, due to differences in vessel dimensions, simulations and experiments will be necessary to determine the optimal combinations for varying parameters in other systems. We have included this information in the manuscript (lines 743-745).

3. Given that this is a lab protocol manuscript, the detailed step-by-step protocol provided in the supplement is appreciated. However, this is likely too much detail for the typical reader. I suggest providing a high-level overview of the protocol in the Methods section of the manuscript so readers have sufficient information to understand the rest of the manuscript without needing to read the lengthy supplement.

RESPONSE: We have included a protocol overview in the Methods section (lines 146-161) to provide readers with sufficient information without requiring review of the supplementary protocol.

4. The viewing chamber described in Expected Results section A is difficult to understand based on a text description. Please provide a schematic figure with dimensions and labels. What is the distance between the TEBV lumen and the microscope objective? What is the “frontal housing lens?” Example images using a light microscope would be informative.

RESPONSE: We have added a new schematic figure (Fig 2A) with dimensions and labels to help understand the text description (Fig 2A caption, line 207-214).

5. Please provide CAD files for the chamber, chamber holder, and mold designs in the supplement. This will allow others to fabricate similar devices.

RESPONSE: All CAD files for the chamber, chamber holder, and mold designs are now included in the supplementary materials (S1 CAD files).

6. In Expected Results section B, it would help to clarify the clamping force is provided by snugly assembling the chamber.

RESPONSE: The TEBV chamber should be tightly closed by hand-tightening the four screws located at the corners of the chamber. Then, use pliers to further tighten the screws evenly until the nuts can no longer be turned. We have clarified that the clamping force is achieved by snugly assembling the chamber components (lines 250-251).

7. Please also express the cell seeding densities as cells per cm2 of seeding area. Given the cells are settling onto a surface, the number per area is more useful than the number per volume.

RESPONSE: In this study, the inner radius of the TEBVs was 0.04 cm, resulting in an estimated inner luminal surface area of 0.25 cm² and a lumen volume of 0.005 cm³ for a 1 cm-long TEBV (lines 526-527). Based on these calculations, we have now reported the EC seeding densities as 1.0×105 (low), 1.5×105 (medium), and 2.0×105 (high) ECs/cm² were utilized (lines 287-288). These cell seeding densities have been incorporated into the revised manuscript.

8. Why was en face imaging not performed to calculate cell coverage within the vessels? Why was sectioning performed and circumferential coverage calculated when the chambers nicely provide the ability to directly visualize the cell coverage on the surface of the vessels (e.g. Figure 8B)?

RESPONSE: En face sectioning often leads to endothelial cell loss during processing due to the small size of the vessels. Additionally, the opaqueness of the collagen layer in our viewing chamber prevents accurate evaluation of endothelial coverage in situ. Therefore, sectioning and circumferential coverage calculations were deemed the most efficient and reliable methods to evaluate endothelial coverage across the vessel length.

9. Why is there no static group for the data presented in Figure 5C-D?

RESPONSE: In Figure 5, we examined EC coverage after 24 hours of perfusion (Fig 5B) and 7 days of perfusion (Fig 5D). Since the static experiments did not undergo any perfusion, the static attachment conditions alone are not a suitable reference group and were therefore removed. The study in Fig 5C-D emphasizes long-term EC coverage under flow. Due to the oxygen and nutrient supply provided by flowing media, static culture of TEBVs for 7 days would result in substantial cell loss and thus was not included in this figure. Furthermore, dynamic perfusion more closely mimics the physiological conditions of blood vessels, aligning with the setup required for downstream applications.

10. Line 346 may have a typo: starting position in Figure 7A appears to be 90 deg.

RESPONSE: We have corrected the typo.

11. It is stated that rotational speeds above 29 rph prevent cells from contacting the vessel wall. Can these data be shown? E.g. 30 rph?

RESPONSE: We have included an additional figure (Fig 7D) to show cell trajectory under a rotational speed of 30 rph.

12. The last section in Expected Results section C describing the forces acting on the cells reads as disconnected from the rest of the manuscript. Please relate this section back to optimal seedings conditions, especially rotational speed.

RESPONSE: We have revised the manuscript to better relate this section to optimal seeding conditions, particularly rotational speed (line 574-576).

13. Please define the white arrows in the caption for Figure 10A.

RESPONSE: In the revised caption for Figure 10A (line 652), we have clarified that the white arrows indicate adhered monocytes on the endothelial cell surface within the TEBV lumen. This has been clarified in the figure caption.

14. In Figure 1A, consider showing hours/minutes rather than days on the timeline.

RESPONSE: We have modified the timeline in the Figure 1A.

15. In Figure 7, consider highlighting the inner wall located at 0.04 cm.

RESPONSE: We have modified Figure 7 to highlight the inner wall location at 0.04 cm.

Reviewer #2:

This study offers an efficient protocol in tissue-engineered blood vessel modeling through the ability to monitor multiple TEBVs simultaneously, developing a viewing chamber for real-time visualization during endothelialization process, simulating the physiological conditions by gradually ramping shear stress, leading to a more stable endothelial layer, demonstrating an optimized process for seeding density and rotation duration, addressing previous gaps in EC adhesion studies and incorporating functional markers (e.g., VE-cadherin, ICAM-1) to confirm endothelial stability and functionality.

All these provide an accessible workflow for vascular research.

So, I recommend this paper for publication in PLOS One journal.

RESPONSE: We are grateful for the reviewer’s recommendation for publication and their acknowledgment of the contributions our protocol offers to vascular research.

In conclusion, we hope these changes adequately respond to the reviewers’ concerns and make the manuscript significantly better. Thank you for your consideration. We look forward to hearing from you.

---

## [Decision Letter · Decision Letter 1]

14 Mar 2025

PONE-D-24-55852R1Strategies for improved endothelial cell adhesion in microphysiological vascular model systemsPLOS ONE

Dear Dr. Truskey, 

Thank you for submitting your manuscript to PLOS ONE. After careful consideration, we feel that it has merit but does not fully meet PLOS ONE’s publication criteria as it currently stands. Therefore, we invite you to submit a revised version of the manuscript that addresses the points raised during the review process.

We look forward to receiving your revised manuscript.

Kind regards,

Ahmed El-Fiqi, Ph.D.

Academic Editor

PLOS ONE

Journal Requirements:

Additional Editor Comments (if provided):

Please, address these two remaining minor issues raised by reviewer #1 prior to decision processing 

Reviewers' comments:

Reviewer's Responses to Questions

**Comments to the Author**

1. Does the manuscript report a protocol which is of utility to the research community and adds value to the published literature?

Reviewer #1: Yes

2. Has the protocol been described in sufficient detail?

To answer this question, please click the link to protocols.io in the Materials and Methods section of the manuscript (if a link has been provided) or consult the step-by-step protocol in the Supporting Information files.

The step-by-step protocol should contain sufficient detail for another researcher to be able to reproduce all experiments and analyses.

Reviewer #1: Yes

3. Does the protocol describe a validated method?

Reviewer #1: Yes

4. If the manuscript contains new data, have the authors made this data fully available?

Reviewer #1: No

**5. Is the article presented in an intelligible fashion and written in standard English?**

Reviewer #1: Yes

6. Review Comments to the Author

Reviewer #1: Two remaining minor concerns:

1. It is still not clear why the average rotation speed is stated to be 24 rph and the range is stated to be 18-22 rph. How can it be the average speed is not within the range of speeds?

2. It is stated the raw data would be provided in a supplementary file entitled "Raw Data." However, there is no reference to this file in the text of the manuscript and it is not provided as a supplementary file.

7. PLOS authors have the option to publish the peer review history of their article (what does this mean? ). If published, this will include your full peer review and any attached files.

**Do you want your identity to be public for this peer review?** For information about this choice, including consent withdrawal, please see our Privacy Policy .

Reviewer #1: No

---

## [Author Response · Author response to Decision Letter 2]

1 Apr 2025

We thank the reviewer for providing additional feedback. Please find below our point-by-point response to the reviewers. The line numbers referenced here correspond to those in the clean manuscript.

1. It is still not clear why the average rotation speed is stated to be 24 rph and the range is stated to be 18-22 rph. How can it be the average speed is not within the range of speeds?

We appreciate the opportunity to further clarify this point. Based on our prior work and the theoretical evaluation of rotational speed during endothelial cell seeding, the initial target range for the rotator system was ~18–22 revolutions per hour (rph). The original rotator, prior to modification, operated at ~ 8 revolutions per minute (rpm).

Upon consultation with our machinist, we determined that the motor supplied with the original rotator was not sufficiently powerful to support the necessary gearing changes to achieve the target speed. Reaching the desired 18–22 rph would have required both replacing the motor and manufacturing a custom worm gear set.

As an alternative, a standard 20:1 worm gear set was available, which reduced the speed to approximately 24 rph. After discussions with our machinist and other team members, we elected to proceed with this readily available option and retain the original motor. The system was then assembled using a custom housing integrated with the rotating assembly. We have included both values in the manuscript to provide practical references for others who may wish to build a similar system and clarified the reasoning behind the discrepancy (line 113-120). To ensure consistency, we addressed this issue in other relevant sections as needed (line 247-248, line 362, line 423-424).

2. It is stated the raw data would be provided in a supplementary file entitled "Raw Data." However, there is no reference to this file in the text of the manuscript and it is not provided as a supplementary file.

Thank you for bringing this issue to our attention. We have included the raw data as the supplementary file. We have also added statement in the figure legends (line 284-285, line 336, line 456-457, line 481-482) to guide readers to the supplementary file entitled "Raw Data."

In conclusion, we appreciate the reviewer’s attention to these issues and hope our explanation resolves the confusion.

---

## [Editor Report · Decision Letter 2]

3 Apr 2025

Strategies for improved endothelial cell adhesion in microphysiological vascular model systems

PONE-D-24-55852R2

Dear Dr. Truskey,

We’re pleased to inform you that your manuscript has been judged scientifically suitable for publication and will be formally accepted for publication once it meets all outstanding technical requirements.

Kind regards,

Ahmed El-Fiqi, Ph.D.

Academic Editor

PLOS ONE

Additional Editor Comments (optional):

Minor issues have been properly addressed

---

## [Editor Report · Acceptance letter]

PONE-D-24-55852R2

PLOS ONE

Dear Dr. Truskey,

I'm pleased to inform you that your manuscript has been deemed suitable for publication in PLOS ONE. Congratulations! Your manuscript is now being handed over to our production team.

Kind regards,

on behalf of

Dr. Ahmed El-Fiqi

Academic Editor

PLOS ONE